# Engaging Live Video Comments Generation

**Figure 1: An example of live video comments generation. Our method aims to generate meaningful and engaging comments (in purple italic) based on video frames, subtitles, and human-posted comments with "like" counts, rather than common expressions such as "hahaha" or "interesting".**

## ABSTRACT

Automatic live commenting is increasingly acknowledged as a crucial strategy for improving viewer interaction. However, current methods overlook the significance of creating engaging comments. Engaging comments can not only attract viewers' widespread attention, earning numerous "likes", but also further promote subsequent social comment interactions. In this paper, we introduce a novel framework for generating engaging live video comments, aiming to resonate with viewers and enhance the viewing experience. Then, we design a Competitive Context Selection Strategy to accelerate differential learning by constructing relatively attention sample pairs with different levels of attractiveness. This approach addresses the sample imbalance problem between highly-liked and low-liked comments, as well as the relative attractiveness issue of comments within video scenes. Moreover, we develop a Semantic Gap Contrastive Loss to minimize the distance between generated comments and higher-liked comments within the segment, while also widening the gap with lower-liked or unliked comments. This loss function helps the model to generate more engaging comments. To support our proposed generation task, we construct a video comment dataset with "like" information, containing 180,000 comments and their "like" counts. Extensive experiments indicate that the comments generated by our method are highly engaging, more fluent, natural, and diverse compared to baselines.

## CCS CONCEPTS

• **Computing methodologies → Natural language generation**; *Computer vision tasks.*

## KEYWORDS

Natural Language Generation, Social Media Processing, Other

## 1 INTRODUCTION

Online video platforms have emerged as the main sources for individuals to access entertainment and various forms of information. Live video comments, also referred to as "bullet screen comments", have increasingly become a distinctive interactive feature on several of these platforms. Unlike normal comments posted after watching a video, the distinguishing feature of live video comments is that they often represent instantaneous reactions and feelings from viewers at a specific timestamp or clip during a video. As shown in Figure 1, the scrolling comments on the right side mainly involve discussions among viewers about the scene at the 31st-minute timestamp of the video, where the two characters are holding their phones. Each "like" received after a comment indicates appreciation from the audience. Some engaging comments stand out among the many ones in the video clip, resonating with lots of viewers, drawing most people's attention, and consequently receiving many "likes". These comments not only greatly enhance the viewing experience for users, but also stimulate further social interactions. As a result, they can finally boost the video's total views, user watch

time, and video popularity. However, existing video comment generation methods lack exploration into generating engaging comments [11, 18, 20, 24].

Generating engaging live video comments faces numerous challenges, but we consider the following three aspects particularly significant: (1) The factors that make a comment appealing are diverse, and approval through likes is inherently subjective. A concise summary of the plot, a well-written description, or an insightful response to other comments can all attract viewers' approval. Therefore, it is challenging to conclusively determine what kind of comments can consistently draw viewers' attention and receive considerable "likes". (2) There exist many general comments that are attractive in specific scenarios. Among the many live video comments, there exist many universal expressions that can attract user interactions and "likes" when they appear in specific video scenarios. For instance, "Looking forward to the next time" is a commonly appreciated sentiment. However, learning solely from these comments would compromise the applicability of the model and its generative diversity. (3) Highly-liked comments are scarce. Whether within specific sections or throughout the entire video, highly-liked comments are limited in number. How to effectively leverage this sparse highly-liked feedback along with numerous barely-liked comments for generation is a key point of this paper.

To address the above challenges, we introduce a multimodal generation framework to accomplish the novel task of creating engaging live video comments. Firstly, we present a Competitive Context Selection approach aimed at tackling the sample imbalance issue between highly-liked and low-liked comments, as well as addressing the relative attractiveness problem of comments within video scenes. It constructs relatively attention sample pairs with varying levels of attractiveness by utilizing the rare highly-liked comments and numerous barely-liked comments in the video clips, competitively determining the division of high and low context based on normalized "likes" scores rather than an absolute fixed threshold. Moreover, we design a Semantic Gap Contrastive Loss that pulls generated comments closer to higher-liked contextual comments and pushes them further from lower-liked ones. This enables the model to learn how to generate highly engaging comments in a given video scenario. To support this engaging comment generation task, we construct a multimodal live video comment dataset from the iQIYI[1] video platform, containing over 180,000 comments, sub-scene video imagery, and subtitles strictly aligned with the video timeline. Unlike previous methods, each comment in our dataset includes actual "like" counts. We aim to leverage this unique data to guide the model in producing natural, relevant, and engaging comments.

Our main contributions are as follows:

- We propose a novel multimodal framework for high-quality and engaging live video comments generation by utilizing the "like" feedback from viewers.
- We develop a Competitive Contextual Selection Strategy to tackle the sample imbalance issue and relative attractiveness problem of comments by constructing relative attention sample pairs containing comments with different levels of "likes".

- We propose a Semantic Gap Contrastive Loss that encourages generated comments closer to higher-liked comments while distancing them from lower-liked comments, aiming for more engaging content generation.
- We construct a multimodal live video comments dataset containing 180k comments accompanied by "like" counts. Extensive experiments validate that our proposed approach outperforms baselines, resulting in engaging, fluent, diverse comments that align with the video clips.

## 2 RELATED WORK

**Multimodal Generation** . Multimodal generation has become a significant research area in artificial intelligence in recent years. Relevant studies include image and video captioning[1, 4, 5, 17], multimodal dialogue generation[2, 15], image and video question-answering[7, 9, 23], and so forth. Video comment generation also stands as an application of multimodal generation in online video sites. Video commenting can be addressed as a downstream task using classical multimodal pre-trained models.

**Video Comments Generation**. Existing works on video comments generation primarily focus on producing fluent, anthropomorphic, and video content-congruent comments. Livebot [11] stands out as a representative masterpiece and is the pioneer in exploring automatic video commenting. It presents two effective video comment generation models and establishes a large-scale video comment dataset for crafting contextually consistent and fluent live video comments. VideoIC [18] is another commendable bullet screen comment generation work, which incorporates a time prediction sub-task to grasp the temporal relationship between multimodal information. Wu et al. introduces a keyword position prediction module to predict the position of keywords and the ending of the comment. These studies integrate multimodal information, producing fluent and meaningful video comments. However, these approaches overlook the role of comment engagement in enhancing the viewers' video-watching experience and promoting social interaction within the video.

## 3 DATASET

To support the task of generating engaging live video comments, we construct a large-scale dataset comprising over 180,276 comments, 78,909 image frames, 7,769 subtitles, and 7,769 audio records. Distinct from previous datasets in related works, each comment in our dataset is annotated with a "like" count for engagement generation.

### 3.1 Data Processing

We collect a wealth of videos and relevant data, including live video comments, subtitles, video timestamps, and audio information from the online video site iQIYI. Then, we process the data in the following steps: 1) Comment filtering: We first exclude comments with sensitive or inappropriate content to avoid the model learning such content. 2) Episode-based Sample Creation: We craft samples based on subtitle start/end timestamps, considering platform transmission delay and human response time. Each sample consists of subtitle information extracted from images within the video clip, along with surrounding comments. 3) "Like" Counts Normalization: The significant disparities in the number of "likes" for the comments will

---

[1]https://www.iqiyi.com/

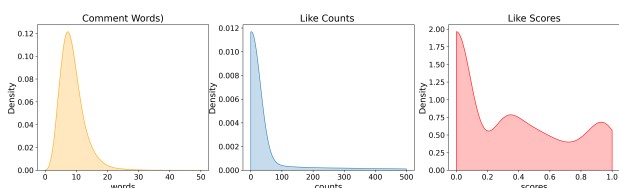

**Figure 2: The proposed engaging live video comment generation framework. It takes video frames, subtitles, and selected contextual comments from the video segment as input, producing fluent and engaging comments. Competitive selection picks comments with 'likes' differences, choosing higher-liked and lower-liked comments as contextual comments for differential learning. $\mathcal{L}_{HCL}$ and $\mathcal{L}_{LCL}$ are employed respectively to narrow the gap with the higher-liked comments and widen the distance from lower ones, bringing the target distribution closer to those relatively engaging comments.**

**Figure 3: Comment words distribution, "likes" count distribution, and normalized "likes" scores distribution.**

**Table 1: Dataset Statistics of Proposed Comments.**

| Item | Value | Item | Value |
|---|---|---|---|
| Comment | 180,276 | Train/Val/Test | 8/1/1 |
| Frame | 78,909 | Train Comment | 144,220 |
| Subtitle&Audio | 7,769 | Val Comment | 18,028 |
| Duration(mins) | 618 | Test Comment | 18,028 |
| Avg Words | 8.76 | Avg Likes | 77.21 |
| Comments/min | 292 | Avg Like Score | 0.32 |

hinder model learning, so we normalize the "like" counts of all comments to facilitate controlled processing. 4) Contextual Comment Selection: Given the scarcity of highly-liked comments, we design a competitive context selection strategy, randomly selecting pairs of comments with normalized scores exceeding a certain value, rather than strictly dividing by a fixed "like" count. 5) Privacy Measures: For user safety, we remove any data potentially identifying the users during processing.

## 3.2 Data Statics

Statistical information in proposed comments dataset is provided in Table 1. In addition to the fundamental video details, we also record

the average number of "likes" and word count, among other things. Figure 3 presents the distribution of word counts, distribution of "like" counts, and distribution of "like" scores after normalization. The long-tail distribution of "likes" signifies the rarity of highly-liked and engaging comments. And after normalization, the data distribution is more even and reasonable.

## 4 METHOD

### 4.1 Engaging Comments Task Definition

Our objective is to generate an engaging live video comment $c_e$, with the generative model $G$, based on the context video information of a specific clip $Context_{t_1,t_2}$:

$$c_e = G(Context_{t_1,t_2}). \qquad (1)$$

The context of the video clip is composed of multimodal video information between two video timestamps $[t_1, t_2]$, primarily including video frames $V_f$, subtitles $T_s$, relatively higher-liked comments $C_h$, and lower ones $C_l$:

$$Context_{t_1,t_2} = \{V_f, T_s, C_h, C_l\}. \qquad (2)$$

Within each clip, apart from the comments $(C_h, C_l)$ selected as part of the conditional context, the remaining comments with high "likes" are extracted to form an engaging reference set $r$, intended for evaluating the appeal of generated comments. It is noteworthy that for each target comment, its higher-liked contextual comment, as well as itself, is not included in the highly-liked reference set $r$.

### 4.2 Engaging Generation Architecture

Figure 2 showcases our proposed framework, comprising five parts: a textual encoder and a visual encoder for subtitle and image feature extraction, a composite decoder to flexibly merge multimodal information and generate comments, a competitive context selection module for differentiated learning, and semantic gap contrastive losses for relative engagement control.

The visual encoder, denoted as $E_V$, accepts the video frame image from a specified timestamp $t_v$ within a video clip $[t_1, t_2]$ to extract video image features. We adopt the structure of ViT [3], where the input video frame image $v_f$ is first partitioned into a patch sequence, and then processed through the encoder to obtain the final frame image representation $e_f$:

$$e_f = E_V(v_f). \tag{3}$$

The text encoder takes subtitle information corresponding to the video frame timestamps as input to extract the subtitle text feature $e_s$. We directly employ the encoder $E_T$ of transformer [16]:

$$e_s = E_T(t_s). \tag{4}$$

Considering that both the context comments and the groundtruth are typically live video comments consistent with the context, sharing the same video subtitle context and having similar linguistic structures, and there is a possibility of interaction or replies among the comments, we concatenate them as the target to enhance the model's training and learning process. The decoder takes the target comment concatenated with contextual comments as its input:

$$c = \{c_{high}, c_{low}, c^*\}, \tag{5}$$

where $c_{high}$ and $c_{low}$ are randomly selected higher-liked and lower-liked video comments from the same video clip ($[t_1, t_2]$), used as the context, and $c^*$ is the groundtruth. The previous output token representation $e_{cpre}$ of the target comment is incorporated with the video frame and video subtitle representations in the decoder. The decoder employs a residual network structure for layer modal feature fusion,

$$out_c = Att(e_{cpre}, e_{cpre}, e_{cpre}), \tag{6}$$

$$out_{cf} = Att(out_c, e_f, e_f) + \gamma_1 out_c, \tag{7}$$

$$out_{cft} = Att(out_{cf}, e_s, e_s) + \gamma_2 out_{cf}, \tag{8}$$

where $\gamma_1, \gamma_2 \in [0, 1]$ are output weight coefficients from the previous layer. The output $out_{cft}$ will be computed through the feed-forward layer to achieve the representation of the current comment token. During the test phase, contextual comments can be provided to the model for the final generation.

### 4.3 Competitive Context Selection Strategy

When constructing the context for the target, we consider the following problems: 1) Highly-liked comments are scarce. As shown in Figure 3, the number of likes for comments follows a long-tail distribution. Even on popular video platforms, a large number of comments may receive few or even no likes. Many scenarios, due to the lack of highly-liked comments, will result in a sharp reduction in training samples, further limiting the model's learning performance and applicability. 2) Engagement is relative. In comments that scroll past in a short time within a clip, besides video and subtitles, comments marked with high likes are more likely to attract viewers' attention. However, human attention is limited. In every video clip, comments like clicking means competition for the viewer's limited attention. We aim to leverage this competitiveness, holding that it isn't solely between highly-liked and barely-liked comments, but can also exist between two highly-liked ones.

Therefore, we introduced a competitive context construction mechanism. For each comment $c_i$, we first utilize the BOX-COX transformation to normalize the number of likes of comments into "like" scores $s$, defined as:

$$s = \begin{cases} \frac{y^\lambda - 1}{\lambda}, \lambda \neq 0 \\ \ln y, \lambda = 0 \end{cases} \tag{9}$$

where $y$ denotes the "like" counts of comments. This transformation also helps to correct the distribution of the overall number of likes, making the "like" data distribution more consistent with the Gaussian distribution and more evenly distributed. For each comment at timestamp $t$, we first randomly select two samples based on normalized like scores from $[t - 5, t + 5]$. If the difference between the normalized scores of the two samples is greater than the threshold value $d$, then the sample with the lower score is determined as a low-attention context sample $c_{low}$, and the sample with the high score is determined as the high-attention context sample $c_{high}$, defined as:

$$c_{con} = \{(c_{high}, s_{high}), (c_{low}, s_{low})\}, \tag{10}$$

$$s_{high} - s_{low} \geq d, \tag{11}$$

where $d$ is the score distance used to regulate the relative engagement discrepancy and we set $d = 0.4$ in this paper. $s_{high}$ and $s_{low}$ represent the scores for the higher-liked and the lower-liked comments.

### 4.4 Semantic Gap Contrastive Loss

The proposed model is trained by Maximum Likelihood Estimation (MLE) to learn the probability distribution $p_\theta(c)$ of generated comments $c$ that align with every video clip context $(v_f, t_s)$, $\theta$ represents the model parameters:

$$\mathcal{L}_{MLE} = -\frac{1}{|c|} \sum_{i=1}^{|c|} \log p_\theta(w_i | c_{<i}, v_f, t_s). \tag{12}$$

$c$ represents the union of the higher-liked comments, lower-liked comments, and the groundtruth, all of which are contextually relevant, denoted as:

$$c = \{w_{h_1}, w_{h_2}..., w_{l_1}, w_{l_2}..., w_{g_1}, w_{g_2}...\}. \tag{13}$$

To enable the model to generate more engaging comments that improve user experience and increase video social popularity, we need to help the model learn to discern the difference between higher-liked comments and lower-liked comments. SimCTG[14] is an effective way to reduce token redundancy in sentences, but just avoiding repetition with barely-liked comments at the token-level is not enough to ensure comment engagement. Considering the proposed engaging comment generation task, contrastive learning should be applied to comments sharing the same video context.

Therefore, we introduce a Semantic Gap Contrastive Loss to make the generated comments closer to higher-liked comments and far away from lower-liked ones. Firstly, we utilize the proposed Competitive Context Selection Strategy to construct positive and negative samples. Then, we design a Higher-Liked Contrastive Loss $\mathcal{L}_{HCL}$ to narrow the semantic gap between the target comment and the higher-liked comment:

$$\mathcal{L}_{HCL} = \max\{\rho_1 - s(h_t, h_h) - s(h_t, h_t), 0\}, \tag{14}$$

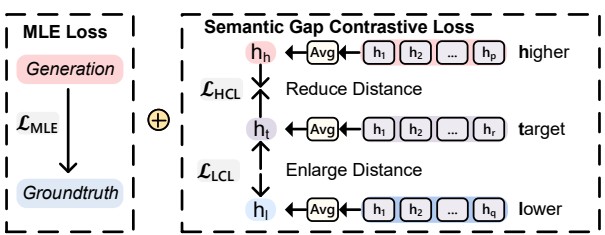

**Figure 4: The final loss function: MLE Loss is responsible for generation quality. The proposed Semantic Gap Contrastive Loss is employed for differentiated learning, driving the model to generate more engaging comments.**

where $s(\cdot)$ denotes the cosine similarity between semantic representations. $h_t$ and $h_h$ are the semantic representations of the target video comment and the higher-liked one respectively. The semantic representation of the comment is obtained by averaging its tokens' hidden states. Correspondingly, we propose Lower-Liked Contrastive Loss $\mathcal{L}_{LCL}$ to enlarge the semantic distance between the target and the lower-liked comment:

$$\mathcal{L}_{LCL} = \max\left\{\rho_2 + s\left(h_t, h_l\right) - s\left(h_t, h_t\right), 0\right\}, \quad (15)$$

where $h_l$ is the representation of lower-liked comments. Apart from the above isolated loss $\mathcal{L}_{HCL}$ and $\mathcal{L}_{LCL}$, we also proposed a Joint Contrastive Loss $\mathcal{L}_{JCL_1}$:

$$\mathcal{L}_{JCL_1} = \max\left\{gap_j, 0\right\}, \quad (16)$$

$$gap_j = \rho_3 + s\left(h_t, h_l\right) - s\left(h_t, h_h\right) - s\left(h_t, h_t\right), \quad (17)$$

Additionally, influenced by the classic contrastive learning loss, we design another joint loss function $\mathcal{L}_{JCL_2}$ for exploration:

$$\mathcal{L}_{JCL_2} = -\log\frac{e^{s(h_t,h_t)/\tau} + e^{s(h_t,h_h)/\tau}}{e^{s(h_t,h_t)/\tau} + e^{s(h_t,h_h)/\tau} + e^{s(h_t,h_l)/\tau}}, \quad (18)$$

where $\tau$ represent a hyperparameter.

Although higher engagement is the keypoint of our task, fluency remains fundamental for comments. Therefore, our final losses for the isolated and joint semantic gap ($\mathcal{L}_{ISG}$ and $\mathcal{L}_{JSG}$) are denoted as:

$$\mathcal{L}_{ISG} = \mathcal{L}_{MLE} + \mathcal{L}_{HCL} + \mathcal{L}_{LCL}, \quad (19)$$

$$\mathcal{L}_{JSG} = \mathcal{L}_{MLE} + \mathcal{L}_{JCL}. \quad (20)$$

As shown in Figure4, MLE pushes the model to closely align with the groundtruth, avoiding context-independent and low-quality outputs. The hidden state of each sentence is obtained by averaging the hidden states of each token in the sentence. Then we apply Semantic Gap Contrastive Loss to make the generation closer to higher-liked comments and enlarge the distance between generated comments and lower-liked comments.

## 5 EXPERIMENT

### 5.1 Implementation Detail

For the text encoder $E_T$ and the visual encoder $E_V$, we utilize the RoBERTa [10] model and ViT [3] structure to extract subtitle features and video frames representations respectively. More detailed

**Table 2: Hyperparameter settings of our model.**

| Hyperparameter | Value |
|---|---|
| hidden state size | 512 |
| optimizer | AdamW |
| batchsize | 128 |
| learning rate | $3\times10^{-4}$ |
| modal weight $\gamma_1, \gamma_2$ | 0.11, 0.11 |
| loss gap $\rho_1, \rho_2, \rho_3$ | 1.5, 0.5, 1.6 |

settings of the encoders can be found in the pre-trained Chinese-CLIP [22]. We employ PyTorch [13] and the transformers library [19] to effectively implement the proposed engaging live video comments generation framework. Hyperparameter settings are presented in Table 2. Four NVIDIA 3090 GPUs are used to train the model for 3 days. $\tau = 0.05$.

### 5.2 Baselines

We compare our engaging live video comments generation method with the following methods:

- **Livebot**: An excellent method for live video comments generation, we employ its transformer version for generation. [11]
- **Livebot$_{clip}$**: A joint approach utilizing CLIP to enhance text-visual features extraction, can be viewed as an enhanced version of Livebot.
- **Cvt**: A transformer visual encoder that retains the advantages of convolutional networks, used for augmenting video visual feature extraction. [21]
- **Vilt**: A classic visual-textual joint multi-modal model, perfectly fitting the proposed comment generation task. [6]
- **Vilt$_{clip}$**: ViLT merges features extracted by CLIP, constituting a strengthened version of ViLT from both textual and visual modalities.

For fairness, all baselines and the proposed approach will employ identical comments context and groundtruth.

### 5.3 Evaluation Metrics

We evaluated the models with the following automatic metrics: **(1) Engagement ($B_r$, $R_r$).** We first construct a reference set for each sample for experimental metric calculation. For each video segment, we collect 10 comments with relatively high likes, serving as the high-attraction reference set. It is noteworthy that these references have no overlap with the contextual comments and groundtruth. We utilize BLEU[12] and Recall metrics to measure the similarity in expression between the generated comments and high-attraction references, represented by $B_r$ and $R_r$ scores respectively. **Semantic Engagement ($Bs_r$).** We employ BertSim to assess the semantic similarity between the generated comments and the high-attraction reference set, thus measuring the proximity of the generated comments to these highly-liked comments in terms of meaning. **(2) Perplexity (PPL).** It is used to evaluate the fluency of generated comments. **(3) Quality (Bertscore[25]: P, R, F1).** The groundtruths selected during samples construction are also

**Table 3: Generation evaluation of ours compared with baselines. (↑: higher is better, ↓: lower is better).**

| Model | Engagement | | | Fluency | Quality | | | Diversity | | | |
|---|---|---|---|---|---|---|---|---|---|---|---|
| | $B_r\uparrow$ | $R_r\uparrow$ | $Bs_r\uparrow$ | PPL↓ | P↑ | R↑ | F1↑ | D1↑ | D2↑ | D3↑ | D4↑ |
| Livebot | 26.19 | 18.27 | 69.39 | 2.38 | 55.53 | 55.56 | 55.48 | 1.59 | 22.29 | 48.80 | 64.18 |
| Livebot$_{clip}$ | 27.10 | 18.75 | 69.51 | 2.11 | 55.48 | 55.57 | 55.46 | 1.60 | 22.87 | 49.59 | 64.89 |
| Cvt | 25.68 | 17.52 | 69.42 | 2.48 | 55.37 | 55.47 | 55.36 | 1.55 | 22.03 | 48.34 | 63.61 |
| Vilt | 25.51 | 17.82 | 69.27 | 1.96 | 55.49 | 55.41 | 55.38 | 1.66 | 24.30 | 52.01 | 66.78 |
| Vilt$_{clip}$ | 24.98 | 16.98 | 69.34 | 2.45 | 55.33 | 55.35 | 55.28 | 1.57 | 23.04 | 50.26 | 65.35 |
| Ours | **41.48** | **33.01** | **70.39** | **1.49** | **57.47** | **57.41** | **57.38** | **1.68** | **24.68** | **52.90** | **68.35** |

**Table 4: Ablation study of our method. Training with only $\mathcal{L}_{MLE}$ means removing $\mathcal{L}_{HCL}$ and $\mathcal{L}_{LCL}$ in $\mathcal{L}_{ISG}$, ablating $\mathcal{L}_{JCL_1}$ in $\mathcal{L}_{JSG_1}$, and removing $\mathcal{L}_{JCL_2}$ in $\mathcal{L}_{JSG_2}$.**

| Loss | Engagement | | | Fluency | Quality | | | Diversity | | | |
|---|---|---|---|---|---|---|---|---|---|---|---|
| | $B_r\uparrow$ | $R_r\uparrow$ | $Bs_r\uparrow$ | PPL↓ | P↑ | R↑ | F1↑ | D1↑ | D2↑ | D3↑ | D4↑ |
| $\mathcal{L}_{ISG}$ | **41.48** | **33.01** | **70.39** | **1.49** | **57.47** | **57.41** | **57.38** | **1.68** | **24.68** | **52.9** | 68.35 |
| $\mathcal{L}_{JSG_1}$ | 39.91 | 31.16 | 70.38 | 1.65 | 57.13 | 57.05 | 57.03 | 1.65 | 23.86 | 51.82 | 67.80 |
| $\mathcal{L}_{JSG_2}$ | 26.24 | 18.07 | 69.43 | 2.08 | 55.56 | 55.55 | 55.49 | 1.62 | 23.68 | 51.26 | 66.98 |
| $\mathcal{L}_{MLE}$ | 39.48 | 30.49 | 70.24 | 1.64 | 57.10 | 56.92 | 56.94 | **1.68** | 24.11 | 52.34 | **68.61** |

comments with relatively higher likes. Besides being more engaging, this implies that these comments are natural, of high quality, and consistent with the video context. Therefore, we use precision, recall, and f1-score of Bertsocre to evaluate the quality of the generated comments. **(4) Diversity[8] (D1, D2, D3, D4).** We use distinct scores to measure the generation diversity.

## 5.4 Evaluation Results

Table 3 showcases the performance of the baseline and our proposed method for generating live video comments. Our method excels in engagement, fluency, quality, and diversity. The higher $B_r$, $R_r$ scores indicate that the comments generated by our model are closer to the higher-liked comments in the expression. The Bertsim score $B_{s_r}$ on the higher-liked reference set $r$ surpasses all baselines by over 0.8, suggesting that our generated comments are semantically closer to the higher-liked comments. Two possible explanations for the lower-than-anticipated increase in semantic similarity scores are: 1) Some comments with fewer likes may share similar meaning with those having more likes, but for various reasons, they did not receive a higher number of likes. 2) In the higher-liked reference set, comments with an extremely high number of likes are rare, and some comments with moderate likes show limited differences from those with few or no likes. The lower PPL suggests that our generated comments are more fluent. A higher Berscore indicates that our generated comments closely resemble the groundtruths, which themselves are higher-liked comments. Our method performs better in Distinct value than baselines, signifying its ability to produce diverse comments. Additionally, apart from the high-attraction reference set, we also construct a low-attraction reference set for each sample by collecting 10 lower-liked comments within the same

context. Then, we further visualize the relative engagement metric results in Figure5. Relative engagement is the ratio of comparing the generated comments with the high-attention reference set and the low-attention reference set respectively after calculating the BELU value.

## 5.5 Ablation Study

While the quality of video comments is influenced by the multimodal information of videos, modality research is not the focus of this paper. We concentrate on leveraging the distinction between highly-liked and less-liked comments for differential learning by our proposed Semantic Gap Contrastive Loss. Table 4 presents the results by various loss functions and pure MLE ablation. When we training the model only by $\mathcal{L}_{MLE}$, it means that we perform ablation experiments on our proposed contrastive losses $\mathcal{L}_{ISG}$, $\mathcal{L}_{JSG_1}$, and $\mathcal{L}_{JSG_2}$. Specifically, for $\mathcal{L}_{ISG}$, it means we remove $\mathcal{L}_{HCL}$ and $\mathcal{L}_{LCL}$. And for $\mathcal{L}_{JSG_1}$ and $\mathcal{L}_{JSG_2}$, it means we ablate $\mathcal{L}_{JCL_1}$ and $\mathcal{L}_{JCL_2}$ respectively.

From the results in the table, we can see that $\mathcal{L}_{ISG}$ performs best and removing the proposed $\mathcal{L}_{HCL}$ and $\mathcal{L}_{LCL}$ will Weaken the performance of the model on various indicators. $\mathcal{L}_{JSG_1}$ performs better in Engagement and Quality values than only using MLE Loss. It means that this joint semantic gap contrastive loss function $\mathcal{L}_{JSG_1}$ has a certain effect on some indicators, but it does not perform as well as $\mathcal{L}_{ISG}$. Surprisingly, $\mathcal{L}_{JSG_2}$ influenced by the classic contrastive learning loss design performs worse than MLE Loss. MLE performs exceptionally well. The joint semantic gap contrastive loss is relatively weaker but still maintain an advantage over baseline methods.

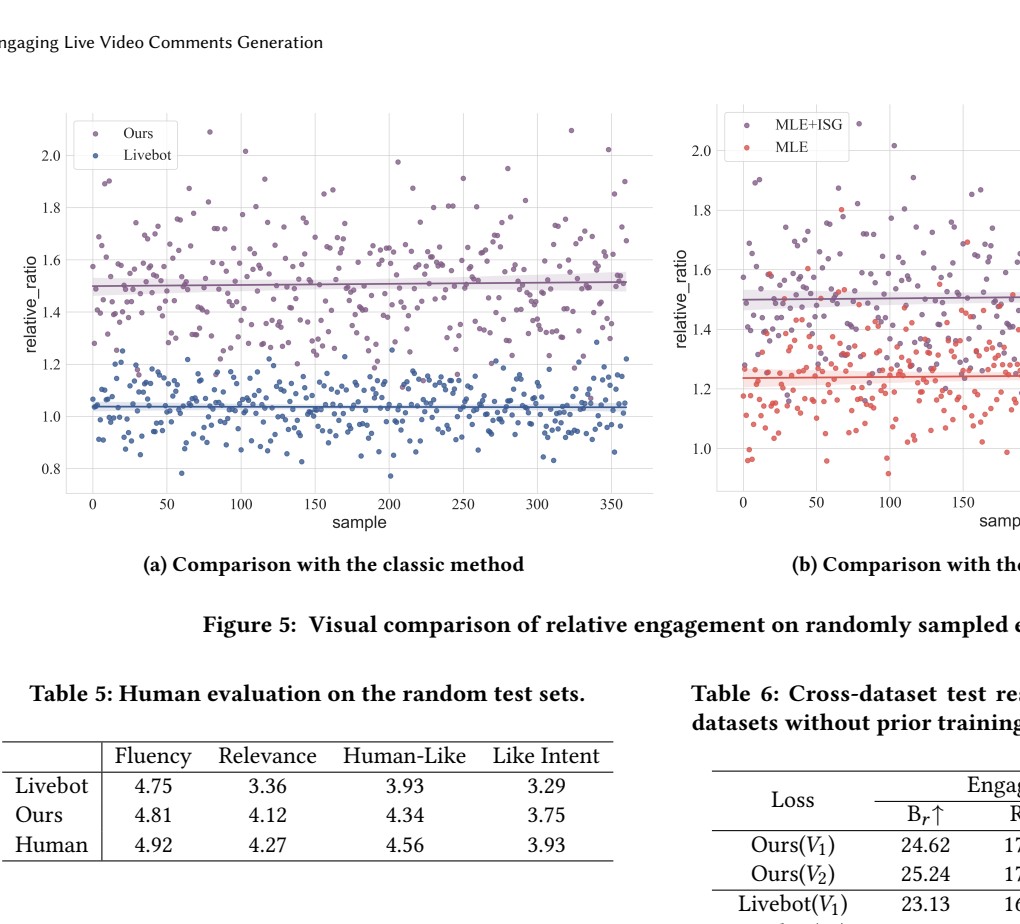

(a) Comparison with the classic method

(b) Comparison with the different losses

Figure 5: Visual comparison of relative engagement on randomly sampled examples

Table 5: Human evaluation on the random test sets.

|  | Fluency | Relevance | Human-Like | Like Intent |
|---|---|---|---|---|
| Livebot | 4.75 | 3.36 | 3.93 | 3.29 |
| Ours | 4.81 | 4.12 | 4.34 | 3.75 |
| Human | 4.92 | 4.27 | 4.56 | 3.93 |

Table 6: Cross-dataset test results on two additional sub-datasets without prior training.

| Loss | Engagement | | | Fluency |
|---|---|---|---|---|
|  | $B_r\uparrow$ | $R_r\uparrow$ | $Bs_r\uparrow$ | PPL$\downarrow$ |
| Ours($V_1$) | 24.62 | 17.03 | 70.11 | 1.57 |
| Ours($V_2$) | 25.24 | 17.21 | 69.37 | 1.57 |
| Livebot($V_1$) | 23.13 | 16.64 | 69.07 | 2.39 |
| Livebot($V_2$) | 23.24 | 16.05 | 68.98 | 2.40 |

## 5.6 Human Evaluation

**Metrics**. We evaluate the models using the following human metrics: 1) **Fluency**: assessing the fluency of live video comments. 2) **Relevance**: determining the relevance between comments and video scene. 3) **Human-Like**: examining the probability that the comments were sent by humans. 4) **Like Intent**: evaluating the engagement of the comments to human. We randomly sampled 100 samples. We employ three individual annotators to score the comments on a scale of [1, 5] across the four aspects.

**Results**. Table 5 displays the results from human evaluations, indicating that our method outperforms the compared baselines and approaches the performance of human-generated comments. It can be observed that the baseline can generate fluent live video comments, but the relevance and human-like scores are weak. As we all know, even if the comments are fluent and general in nature, they may also struggle to draw viewers' attention and interaction if they do not closely match the context of the video clip. Our model achieves better performance in these four aspects than baselines especially in Relevance, Human-Like, and Like Intent scores, which means comments generated by our model are more appealing to viewers.

## 5.7 Cross-Dataset Study

We conduct cross-dataset experiments to evaluate the generalization performance of our proposed method. Training solely on a single dataset might yield excellent performance due to overfitting and learning ample contextual scenes. However, such a model might excel in specific video types but struggle to adapt to others. Apart

from the dataset primarily used for training and testing in this study, we also conduct untrained cross-dataset testing on two distinct datasets, $V_1$ and $V_2$, by randomly extracting 10k samples from each. $V_1$ comprises 140k comments, 64k images, and 5k subtitles, while $V_1$ consists of 241k comments, 111k images, and 11k subtitles. Table 6 presents the cross-dataset testing results without prior training. The relatively stable PPL scores suggest that the model can still generate fluent video comments. It is observed that the best score decreases by roughly 15%, with a more significant drop in the positive reference set and a slight decline in bertsim. It may be because there are still some similar scenes between datasets, and some general but potentially popular comments remain applicable. However, for some sophisticated and interesting comments tailored to new video clip contents (e.g., character names, specific terms, character relationships), the model struggles to learn new expressions without training, leading to a considerable disparity between the highly-liked reference set and the two additional datasets. Additionally, we also report the performance of Livebot in cross-dataset testing in Table 6. It is noticeable that Livebot's Engagement score does not drop much, but still remains less engaging compared to our method. We believe this is mainly because when the model encounters unfamiliar scenes (including different frames, subtitles, and comment contexts), it tends to generate more generic comments that align with some of the generic comments found in highly praised comments.

**Table 7: Generation speeds of different models. A generation time of 0.1s indicates that the model can produce live video comments based on video segments, allowing them to scroll or fly across the screen instantly.**

| Models | Livebot | Livebot$_{clip}$ | Cvt |
|---|---|---|---|
| G-Speed(s) | 0.087 | 0.089 | 0.090 |
| Models | Vilt | Vilt$_{clip}$ | Ours |
| G-Speed(s) | 0.099 | 0.092 | 0.122 |

**Video Frame**

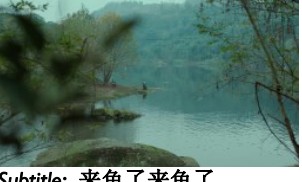

**Subtitle:** 你每次见我第一句话
*Every time you see me,*
*the first thing you say is...*

**Subtitle:** 来鱼了来鱼了
*The fish are coming! The*
*fish are coming!*

**Comments**

*Contextual Comment*
你说你能做成啥？575👍
*What do you think you can achieve?*
每天同一话。2👍
*It's the same thing you say every day.*
*Groundtruth*
你都造成三回了？671👍
*You've done it three times!*
*Generated Comment*
又又又又对不起啊？
*Sorry again and again and again!*

*Contextual Comment*
我宝就是那条鱼？264👍
*My sweet is that fish.*
好位置。0👍
*Ideal Position.*
*Groundtruth*
《等鱼上钩》919👍
*"Waiting for the Fish to Bite".*
*Generated Comment*
上来的可能不是鱼
*What's coming up might not be a fish.*

**Figure 6: Examples generated by our method.**

## 5.8 Comment Generation Speed

For standard videos where comments are superimposed, the generation timing is not constrained; one can produce the comment and then decide on a specific timestamp within the corresponding video segment. However, when deploying real-time comment generation applications, generation speed is crucial, as it determines the model's ability to promptly respond to video clips and ongoing comments. Table 7 lists the average generation time for each model, and an approximate comment generation time of 0.1s is sufficient to craft comments that match the video clip context. Although our proposed model is slightly slower in generating comments compared to previous models, the 0.122s per comment remains within an acceptable range.

## 5.9 Case Study

Figure 6 displays some generated samples (in italic) on our proposed video dataset. Through these examples, it can be observed that our approach can not only generate natural, fluent, and contextually relevant comments but also possesses a certain appeal. For instance, the generated three "again" not only conveys repetition but also reflects emotion in an interesting way. When compared to highly-liked comments, the generated remain competitively engaging.

## 6 CONCLUSION

This paper presents a multimodal framework aiming to generate engaging comments in live video contexts. We build a large-scale multimodal dataset for live video comments where subtitles and timestamps are perfectly synchronized with video timestamps, and each comment is associated with its "likes" count information. Our proposed Competitive Context Selection Strategy helps to construct contrastive learning pairs with higher-liked and lower-liked comments, mitigating the problem of "likes" data imbalance. Moreover, we introduce several Semantic Gap Contrastive Losses to motivate the model to create more appealing live video comments by narrowing the semantic disparity between generated comments and higher-liked comments and enlarging the distance with lower-liked ones. Comprehensive experiments affirm the efficacy of our approach. In future work, we will explore the potential audience for the most-liked comments, deepening our understanding of participation dynamics in live video interactions.

## 7 ETHICAL CONSIDERATIONS

While collecting comment data from online video platforms, we observed that comments often contain metadata, including unique platform account IDs. Although this information is considered public according to platform guidelines, it involves aspects of individual activity and personal privacy that require careful handling. Therefore, for user safety, we remove all data involving private information, keeping only the comment content and its associated video segment timestamp. Additionally, we filter the dataset initially to exclude inappropriate comments. Moreover, It is worth noting that, due to the strict comment moderation rules on existing popular video social platforms, our model is supervised when automatically posting comments. Therefore, our model will be properly monitored and not spread inappropriate or harmful content on social platforms.

## 8 LIMITATIONS

Although we have proposed an effective multimodal framework to generate engaging live video comments in this paper, there are still some limitations. Expanding and incorporating a broader range of video types is essential. Highly-liked comments are often closely related to particular video segments, which may involve some specialized terms or domain-specific object recognition. Without adequate training data or the integration of external knowledge, the expression capacity of the model would be severely limited. Our method utilizes pre-trained Chinese CLIP to enhance feature extraction from video frames and subtitles, but relying too heavily on CLIP's pre-training limits its versatility. Unlike previous comment generation works, we need to consider the quantity and distribution of comments. The scarcity of highly-liked comments limits our ability to collect data rapidly on a large scale. In addition to the dataset used for training and testing, we are actively collecting data from a wider range of video categories, having gathered an additional 500k comments spanning more video types. This expansion paves the way for future research explorations.

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
