# OpenReview forum: "Engaging Live Video Comments Generation"
_acmmm.org/ACMMM/2024/Conference — MM2024 Poster_

### Official Review · Reviewer_Y78t · 2024-05-23

**Rating:** 5
**Confidence:** 3

**Summary:**

This paper introduces a new task called “engaging live video comments generation”, which focuses on generating meaningful comments based on video frames, subtitles and human-posted comments with like counts. They contribute a dataset containing 180,000 comments as well as a model to address this task. The model is a decoder-only transformer with visual and text features. They performed a series of experiments to set up benchmarks for this task.

**Strengths:**

(1) The task is a novel task.
(2) The paper contributes a dataset.
(3) Dataset curation, benchmark setup and experiment design are reasonable and clearly described.

**Limitations:**

(1) Baselines are all before 2022, which are out-of-date.

**Suitability:**

3

---

### Official Review · Reviewer_JjWz · 2024-05-25

**Rating:** 3
**Confidence:** 3

**Summary:**

This paper focus on the task of live comment generation. Specifically, it aims to produce meaningful and engaging comments. To this end, this paper first constructs a large-scale dataset. Then, it propose a Competitive Context Selection Strategy to extract higher-liked and the lower-liked comments. Finally, they design a Semantic Gap Contrastive Loss to train the generation model. Results show that the proposed model outperform several baselines.

**Strengths:**

1. This paper constructs a large-scale dataset, which will make a contribution for the community.

**Limitations:**

1. This paper compares the proposed pertraining based model  with a weaker model Livebot which has not undergone pretraining.
2. The part s(h_t, h_t) is a constant. What the funciton is it?

**Suitability:**

3

---

### Official Review · Reviewer_9qEZ · 2024-05-28

**Rating:** 5
**Confidence:** 2

**Summary:**

This paper introduces a multimodal framework designed to generate engaging comments in live video contexts. The authors develop a multimodal dataset for the live comments, where substitle and timestamps are synchronized with the video. In this paper, they propose a contrastive loss strategy to construct contrastiev learning pairs with higher-liked and lower-liked comments, mitigating the problems of data imbalance. Several experiments and ablation testing are used to demonstrate the efficacy of the proposed method.

**Strengths:**

This paper made use of higher-liked and lower-liked comments combined with video frames to form a multimodal data source. It also conducted extensive experiments to demonstrate the effectivenss of the model.

**Limitations:**

The paper used the Box-Cox transformation to normalize the "like" comments. It would be beneficial if the authors could explain why they chose the Box-Cox transformation over other methods, such as the Yeo-Johnson transformation or the log transformation.

In the ablation testing section, the joint semantic gap contrastive loss function positively impacted some indicators. It would be helpful if the authors could explain why this occurs. Providing an explanation for the ablation testing results is necessary.

**Suitability:**

3

---

### Official Review · Reviewer_itFG · 2024-05-28

**Rating:** 4
**Confidence:** 3

**Summary:**

With the development of live video, live comments have become increasingly important in building a user-attractive live atmosphere and enhancing the interaction experience. Previous work on live video comment generation neglects the engagement of comments. Therefore, this paper studies a new task: engaging live comment generation. With specially designed mechanisms, such as Competitive Context Selection, the model achieves optimal comment generation. The experimental results demonstrate the effectiveness.

**Strengths:**

1. This paper is the first to consider "likes" as a factor in generating more engaging live comments.
2. This paper proposes a live comment dataset and develops a specially designed model to capture the "like" feature for generating engaging live comments.
3. The experimental results of the proposed model outperform previous work.

**Limitations:**

1. It is unclear whether the model accesses a single image or a series of video frames as input for live comment generation. As shown in line 352 on page 4, the input seems to be a single frame. However, according to the description in line 329 on page 3, the input seems to be a short video clip. Given the task requirements, the input should be a short video since live comments need contextual video as a resource rather than a single image. If the author chooses a single frame, how is the most optimal image selected as input?
2. The technical part is somewhat old-school. Currently, multimodal large language models have shown great performance in various multimodal content and generation tasks. As a newly proposed task involving multimodal content and text generation, utilizing MM large language models is indeed a preferred choice. Even without MM large language models, utilizing large language models is also a promising option for text generation. The model with a RoBERTa encoder seems suboptimal for such a generative task. The carefully designed structure, such as HCL and LCL, mostly achieves only trivial improvements, which may be far from the potential benefits of using generative language models.
3. Typos:
   - Page 5, line 495: `),` -> `).`
   - Page 5, line 508: `Figure4` -> `Figure 4`

**Suitability:**

3

---

### Meta-Review · Area_Chair_xryp · 2024-07-03

**Recommendation:** Accept (Poster)
**Confidence:** 5

**Metareview:**

This paper introduces a novel task by considering "likes" as a factor in generating more engaging live comments, marking it as a pioneering effort in this domain. The authors contribute a new live comment dataset and propose a model specifically designed to capture the "like" feature, demonstrating superior performance over existing methods. Despite its strengths, the paper has some limitations. The input format for live comment generation remains ambiguous, with conflicting descriptions suggesting a single image or a short video clip. Additionally, using a RoBERTa encoder may be suboptimal compared to modern multimodal large language models. The choice of the Box-Cox transformation for normalizing "like" comments requires further justification, and the impact of the joint semantic gap contrastive loss function in ablation testing needs a more detailed explanation. Moreover, comparing the outdated Livebot model and pre-2022 baselines indicates room for improvement. Despite these issues, the paper’s contributions to dataset creation, benchmark setup, and experimental design are substantial, making it a valuable addition to the field.